# The DYT6 dystonia causative protein THAP1 is responsible for proteasome activity via *PSMB5* transcriptional regulation

Yan Wang [1,2,7], Yi Wang[1,7], Tomohiro Iriki[1], Eiichi Hashimoto [1], Maki Inami[1], Sota Hashimoto[1], Ayako Watanabe[3], Hiroshi Takano[4], Ryo Motosugi[1], Shoshiro Hirayama[1], Hiroki Sugishita[5,6], Yukiko Gotoh [5,6], Ryoji Yao [4], Jun Hamazaki [1] & Shigeo Murata [1] ✉

The proteasome plays a pivotal role in protein degradation, and its impairment is associated with various pathological conditions, including neurodegenerative diseases. It is well understood that Nrf1 coordinates the induction of all proteasome genes in response to proteasome dysfunction. However, the molecular mechanism regulating the basal expression of the proteasome remains unclear. Here we identify the transcription factor THAP1, the causative gene of DYT6 dystonia, as a regulator of proteasome activity through a genome-wide genetic screen. We demonstrated that THAP1 directly regulates the expression of the *PSMB5* gene, which encodes the central protease subunit β5. Depletion of THAP1 disrupts proteasome assembly, leading to reduced proteasome activity and the accumulation of ubiquitinated proteins. These findings uncover a regulatory mechanism for the proteasome and suggest a potential role for proteasome dysfunction in the pathogenesis of dystonia.

The ubiquitin-proteasome system (UPS) is widely conserved in eukaryotes and is responsible for the degradation of ubiquitinated proteins[1]. This metabolism of proteins via the UPS plays a pivotal role in regulating essential cellular events and maintaining cellular homeostasis through a multitude of processes, including the cell cycle[1], signal transduction[2], DNA repair[3], and protein quality control[4]. A chain of lysine 48-linked ubiquitin molecules is covalently added onto target proteins via ubiquitin-activating enzymes (E1), ubiquitin-conjugating enzymes (E2), and ubiquitin ligases (E3), and serves as the primary signal for recognition by the 26S proteasome[5].

The 26S proteasome is a large, multi-subunit protease complex that degrades ubiquitinated proteins in an ATP-dependent manner. The 26S proteasome consists of a central 20S core particle (CP) capped at one or both ends by 19S regulatory particles (RPs)[6]. The CP, which is

responsible for the proteolytic activity, has a cylindrical structure in which an α-ring, composed of α1–7 subunits, and a β-ring, consisting of β1–7 subunits, coaxially overlap in the order of α-β-β-α[7–9]. The α-ring serves as a gate for entry into the CP of substrate proteins that are unfolded by the RPs. The β1, β2, and β5 subunits, which are associated with caspase-like, trypsin-like, and chymotrypsin-like activities, respectively, confer the ability to cleave peptide bonds and degrade substrate proteins that have entered the CP. The RPs are divided into two subcomplexes, termed the base and the lid. The base consists of six different homologous AAA-ATPase subunits (Rpt1–6) and three non-ATPase subunits (Rpn1, Rpn2, and Rpn13), whereas the lid is composed of nine non-ATPase subunits (Rpn3, Rpn5–9, Rpn11–12, and Rpn15)[10].

Many proteins have been shown to regulate the biogenesis and function of the 26S proteasome. The assembly of the proteasome is

[1]Laboratory of Protein Metabolism, Graduate School of Pharmaceutical Sciences, the University of Tokyo, Bunkyo-ku, Tokyo, Japan. [2]The Affiliated Kangning Hospital of Ningbo University, No. 1 Zhuangyunan Road, Ningbo, China. [3]One-stop Sharing Facility Center for Future Drug Discoveries, Graduate School of Pharmaceutical Sciences, the University of Tokyo, Bunkyo-ku, Tokyo, Japan. [4]Department of Cell Biology, Cancer Institute, Japanese Foundation for Cancer Research, Koto-ku, Tokyo, Japan. [5]Laboratory of Molecular Biology, Graduate School of Pharmaceutical Sciences, the University of Tokyo, Bunkyo-ku, Tokyo, Japan. [6]International Research Center for Neurointelligence (WPI-IRCN), the University of Tokyo, Bunkyo-ku, Tokyo, Japan. [7]These authors contributed equally: Yan Wang, Yi Wang. ✉e-mail: smurata@g.ecc.u-tokyo.ac.jp

mediated by multiple dedicated chaperones, including PAC1/Pba1, PAC2/Pba2, PAC3/Pba3, PAC4/Pba4, and POMP/Ump1 for CP assembly, and S5b/Hsm3, p27/Nas2, p28/Nas6, and PAAF1/Rpn14 for RP assembly[11–13]. In multicellular organisms, the transcription factor Nrf1 and its orthologs orchestrate the coordinated expression of all proteasome subunit genes in response to proteasomal inhibition[14,15]. Given that most genetic approaches employed to study the proteasome have been performed in budding yeast, exploring the mechanism of proteasomal regulation in mammals is imperative, as the mechanisms of proteasomal regulation becomes increasingly diverse and complex in multicellular organisms, and as human diseases associated with proteasome dysfunction are becoming known.

It is widely accepted that proteasomal dysfunction disrupts cellular homeostasis, contributing to the development of various diseases and accelerating aging[16–18]. For example, aged organisms have been reported to exhibit a decline in proteasome activity, resulting in the accumulation of aberrant and potentially deleterious proteins. Moreover, mutations in the proteasome subunits have been shown to be pathogenic. An intronic homozygous variant in PSMC3 and heterozygous de novo nonsense mutations in PSMD12 have been reported to cause neurodevelopmental disorders[19,20]. Similarly, mutations in PSMB8 ($PSMB8^{G197V/G197V}$) and heterozygous variants of PSMB9 ($PSMB9^{G156D/+}$) have been identified in patients with autoinflammatory and immunodeficiency diseases[21–23]. Conversely, studies have shown that enhancing proteasome activity can extend the lifespan of nematodes and fruit flies[24,25]. The overexpression of the β5 subunit extended the lifespan of fruit flies[26]. This is attributed to an increase in the quantity of active proteasomes, as overexpression of the β5 subunit has been shown to increase the amount of assembled proteasomes in human cells, underscoring the importance of β5 in proteasome function[27]. It is evident that the regulation of proteasome function is of significant biological importance. However, the detailed molecular mechanisms underlying proteasome regulation remain incompletely understood.

To gain a comprehensive understanding of the regulatory mechanisms governing proteasome function, a genome-wide CRISPR knockout screen was conducted. This approach identified the transcription factor THAP1 as a regulator of the proteasome. THAP1 is a member of the THAP family of transcription factors and is assumed to be involved in neuronal development, myelination, and DNA repair[28–31]. Several mutations in THAP1 have been linked to DYT6 dystonia, a neurological disorder characterized by involuntary muscle contractions or abnormal postures in movement[32,33]. Nevertheless, it remains unclear which target gene of THAP1 is responsible for the pathogenesis of DYT6 dystonia. We demonstrated that THAP1 knockout resulted in reduced proteasome activity and the accumulation of ubiquitinated proteins due to defects in the assembly of the CP. Further investigation revealed that knockout of THAP1 specifically reduced the expression of the β5 subunit without perturbation of other proteasome subunit expressions. We also showed that THAP1 binds to the promoter region of PSMB5, thereby regulating proteasome activity. These results suggest a potential link between proteasomal dysfunction and the pathogenesis of dystonia.

## Results

### Identification of THAP1 as a regulator of the proteasome through a genome-wide screen

To identify regulators of the proteasome, we employed a genome-wide CRISPR knockout screen. To monitor proteasome dysfunction, we utilized an artificial protein, the mouse ornithine decarboxylase (mODC) degradation domain (amino acids 410–461) fused to the fluorescent protein ZsGreen (ZsGreen-mODC). ZsGreen-mODC is continually degraded following translation under steady conditions and accumulates under proteasome-impaired conditions, exhibiting green fluorescence (Supplementary Fig. 1a). We established U2OS cell lines stably expressing mCherry-P2A-ZsGreen-mODC to monitor cells

with reduced proteasome activity resulting from gene knockout, utilizing mCherry as a reference to normalize construct expression levels. We performed a flow cytometry-based CRISPR knockout screen (Supplementary Fig. 1b). The top 557 genes with a robust rank aggregation (RRA) score less than 0.01, corresponding to -$Log_{10}$ (RRA score) > 2, were defined as positive hits (Fig. 1a). Proteasome subunits were identified as top hits, confirming the reliability of the screen (Fig. 1a and Supplementary Fig. 1c). Within the top 50 genes, THAP1 had a relatively high RRA score, aside from the proteasome subunits, and had not been reported to be associated with the proteasome, making it a likely proteasome regulator (Supplementary Fig. 1c).

To confirm the results of the screen, we expressed two distinct sgRNAs targeting THAP1 in HEK293T and U2OS cell lines expressing mCherry-P2A-ZsGreen-mODC. THAP1 knockout resulted in a significant accumulation of ZsGreen-mODC in both HEK293T (Fig. 1b, c) and U2OS (Fig. 1d, e) cells, comparable to that observed in cells knocked out for the CP subunit α1 encoded by PSMA6. We also measured proteasome activity biochemically and observed a significant reduction in proteasome peptidase activity as a result of THAP1 knockout (Fig. 1f and Supplementary Fig. 1d). To ascertain the impact of THAP1 knockout on proteasome activity, we analyzed cell lysates by immunoblotting. Consistent with the decrease in proteasome activity, THAP1 knockout led to a marked accumulation of ubiquitinated proteins (Fig. 1g). The reduction in proteasome activity and the increase in ubiquitinated proteins resulting from THAP1 knockout were rescued by overexpression of sgRNA-resistant cDNA for THAP1 (Fig. 1h, i, and Supplementary Fig. 1e). These results demonstrate that THAP1 is a proteasome regulator.

### THAP1 is essential for the assembly of the proteasome CP

To understand the mechanism by which THAP1 knockout decreases proteasome activity, cell lysates from THAP1 knockout cells were analyzed by immunoblotting against proteasome subunits and assembly chaperones. We found an accumulation of precursor of β1 and β2 subunits, whose propeptides are cleaved upon completion of CP assembly, and a marked decrease in the other β subunits, β3–7, while α subunits and RP subunits were less affected (Fig. 2a and Supplementary Fig. 2a). Furthermore, we observed an accumulation of CP assembly chaperones PAC2 and Ump1, suggesting the accumulation of intermediates during CP assembly (Fig. 2a). These results suggest a defect in proteasome assembly upon THAP1 knockout.

Previous studies have detailed the process of CP assembly. An α-ring is first formed, onto which β subunits are incorporated in the order of β2, β3, β4, β5, β6, β1, and β7, of which β2, β5, and β1 have propeptides that are cleaved off upon completion of CP assembly (Fig. 2b).

To investigate in detail which step of CP assembly was impaired by THAP1 knockout, cell lysates were fractionated by glycerol gradient centrifugation, and each fraction was subjected to a peptidase activity assay and immunoblotting. THAP1 knockout resulted in a decrease in proteasome activity in both 20S and 26S fractions (Fig. 2c and Supplementary Fig. 2b, c). Consistent with this, CP subunits were decreased in both 20S and 26S fractions of THAP1 knockout (Fig. 2d). The most striking difference was the accumulation of CP assembly intermediates, as evidenced by an increase in the precursor form of β2, β3, and β4 as well as α2 in fractions lighter than the assembled CP (20S) in THAP1 knockout (Fig. 2d, fractions 8–12). The other β subunits, β1, β5, β6, and β7, were barely observed in these fractions; rather, the precursor form of β1 was found in even lighter fractions, presumably in the orphan subunit state without incorporation into CP assembly intermediates (Fig. 2d, fractions 2–6). Considering the assembly process of the CP[34] (Fig. 2b), these results indicate that CP assembly proceeded up to the incorporation of β2, β3, and β4, but the incorporation of β5 and beyond was severely compromised, most likely in the process of β5 incorporation. The absence of the precursor forms of β5 in

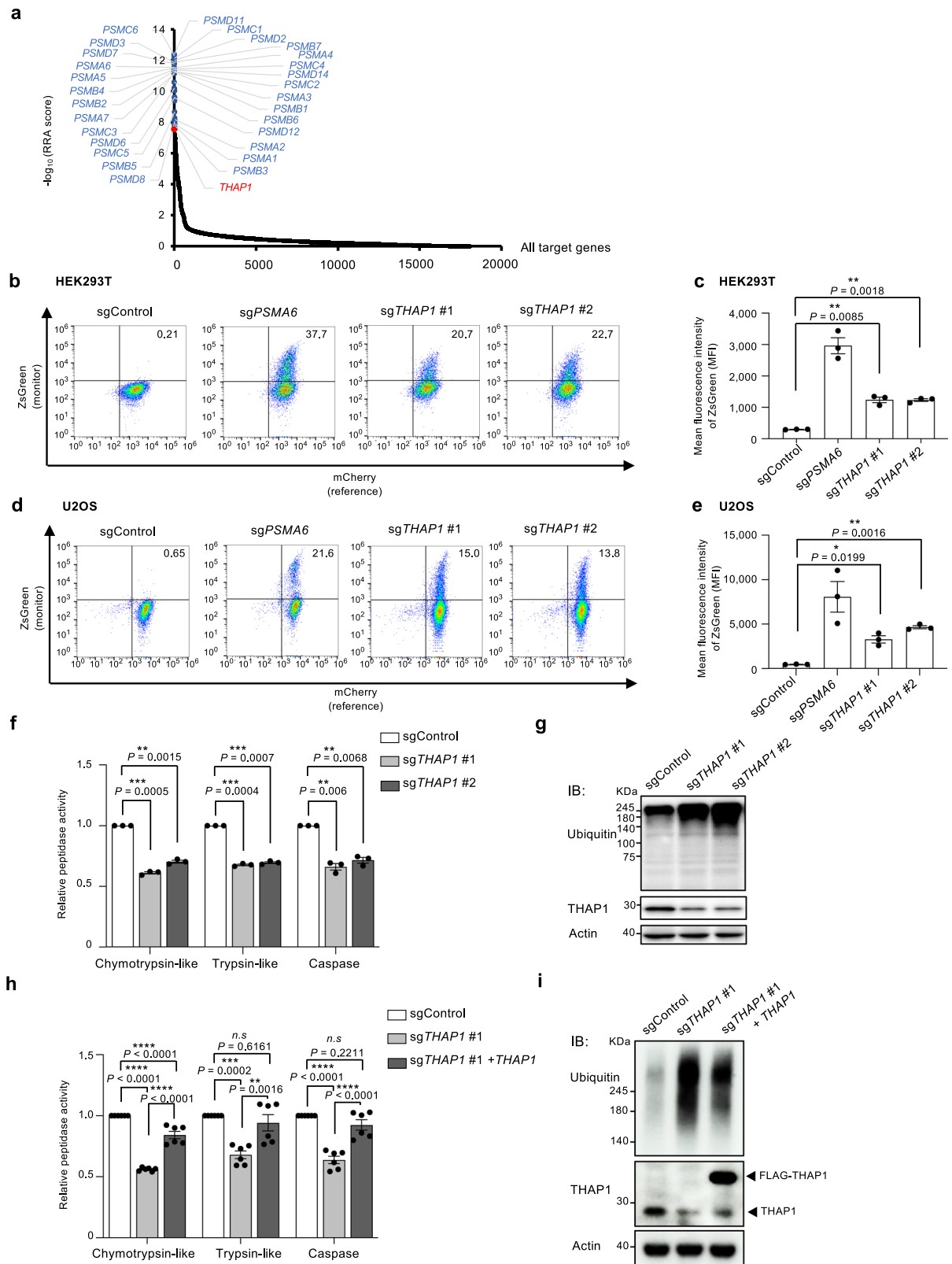

the accumulated assembly intermediates, nor in the even lighter fractions, suggests that β5 expression is rate limiting in *THAP1* knockout cells.

## THAP1 regulates the expression of *PSMB5*

Since THAP1 has been reported to function as a transcription factor with a DNA-binding THAP domain, we reasoned that THAP1 might directly regulate the transcription of the proteasome subunits. We examined the mRNA levels of proteasome subunits and assembly chaperones in *THAP1* knockout cells using RT-qPCR. Of the genes examined, we found that only the mRNA of β5, encoded by *PSMB5*, was significantly decreased in *THAP1* knockout cells, whereas the mRNA levels of other β subunits (Fig. 3a), α subunits, RP subunits, and assembly chaperones were not affected by *THAP1* knockout

**Fig. 1 | Identification of THAP1 as a regulator of the proteasome through a genome-wide screen. a** The dots represent the RRA scores of genes in the CRISPR screen. The blue dots and red dot represent proteasome subunits and *THAP1*, respectively. **b, d** Flow cytometry analysis of HEK293T (**b**) and U2OS cells (**d**) stably expressing mCherry-P2A-ZsGreen-mODC transfected with sgControl (negative control), sg*PSMA6* (positive control), and sg*THAP1*. The numbers in the upper right quadrant of (**b**) and (**d**) represent the percentage of the total population. **c, e** Mean fluorescence of ZsGreen-positive cells in the upper right quadrants of (**b**) and (**d**), respectively. Data represent the mean ± SEM ($n = 3$ from three biological replicates). The significance were calculated using an unpaired two-tailed Student's t-test with Welch's correction and one-way ANOVA Tukey with multiple comparisons test. **f** Proteasome chymotrypsin-, trypsin-, and caspase-like activities of HEK293T cells transfected with sgControl and sg*THAP1*. The activity was measured in the absence of SDS for measuring the 26S activity. **g** HEK293T cells were transfected with the indicated sgRNA and subjected to immunoblot analysis with antibodies against the indicated proteins. **h, i** The chymotrypsin-, trypsin-, and caspase-like activities (**h**) and

immunoblot analysis (**i**) of the *THAP1* knockout cells in which sgRNA-resistant *THAP1* cDNA was added back. The 26S proteasome activities were measured in the absence of SDS. Data represent the mean ± SEM ($n = 3$ from three biological replicates). The significance were calculated using one-way ANOVA Tukey with multiple comparisons test, with exact $P$-values as follows: Chymotrypsin-like relative peptidase activity: ****$P < 0.0001$ ($P_{\text{sgControl vs sg}THAP1\ \#1}$); ****$P < 0.0001$ ($P_{\text{sgControl vs sg}THAP1\ \#1+THAP1}$); ****$P < 0.0001$ ($P_{\text{sg}THAP1\ \#1\ \text{vs sg}THAP1\ \#1+THAP1}$), Trypsin-like relative peptidase activity; **$P < 0.001$ ($P_{\text{sgControl vs sg}THAP1\ \#1} = 0.0002$); *n.s.*, not significant, ($P_{\text{sgControl vs sg}THAP1\ \#1+THAP1} = 0.6161$); **$P < 0.01$ ($P_{\text{sg}THAP1\ \#1\ \text{vs sg}THAP1\ \#1+THAP1} = 0.0016$), Caspase relative peptidase activity: ****$P < 0.0001$ ($P_{\text{sgControl vs sg}THAP1\ \#1}$); *n.s.*, not significant, ($P_{\text{sgControl vs sg}THAP1\ \#1+THAP1} = 0.2211$); ****$P < 0.0001$ ($P_{\text{sg}THAP1\ \#1\ \text{vs sg}THAP1\ \#1+THAP1}$). Data represent the mean ± SEM ($n = 3$ from three biological replicates). *n. s., not significant.* *$P < 0.05$; **$P < 0.01$; ***$P < 0.001$; ****$P < 0.0001$. All experiments were performed at least three biologically independent times with similar results. Source data are provided as a Source Data file.

(Supplementary Fig. 3). The reduction in the mRNA levels of β5 resulting from *THAP1* knockout was rescued by overexpression of sgRNA-resistant cDNA of THAP1 (Fig. 3b). These results suggest that THAP1 specifically regulates the expression of the β5 subunit but not other subunits.

THAP1 has been implicated in the expression of genes involved in neurodevelopment and has not previously been associated with the expression of a proteasome gene. To comprehensively identify the primary transcriptional targets of THAP1, we performed RNA-Seq analysis on *THAP1* knockout cells. We confirmed that among the proteasome subunits, only the expression of *PSMB5* was downregulated in *THAP1* knockout cells (Fig. 3c). *THAP1* knockout also affected the expression of many other genes. Gene ontology analysis of genes downregulated by *THAP1* knockout revealed that THAP1 also regulates genes involved in ribosome biogenesis and mitochondrial function (Fig. 3d and Supplementary Data 1). These results suggest that THAP1 is involved in various cellular homeostasis mechanisms, in addition to proteasome function.

### Overexpression of β5 rescues proteasome dysfunction caused by *THAP1* knockout

To examine whether the regulation of proteasome activity by THAP1 is mediated through the regulation of *PSMB5* expression, we transiently overexpressed each β subunit in *THAP1* knockout cells for five days. Of the β subunits, only the overexpression of β5 significantly rescued proteasome activity (Fig. 4a). The increase in ubiquitinated proteins resulting from *THAP1* knockout was also rescued by overexpression of β5 (Supplementary Fig. 4a). The cell lysates were fractionated by glycerol gradient centrifugation and subjected to peptidase activity assays and immunoblot analysis. Consistently, only β5 overexpression restored proteasome activity in the 20S and 26S fractions (Fig. 4b and Supplementary Fig. 4a, b), reduced the accumulation of orphan β1 (Supplementary Fig. 4b, fractions 2–8) and the precursor β2 (Supplementary Fig. 4b, fractions 10–14), and increased the amount of assembled proteasomes (Supplementary Fig. 4b, fractions 22–26).

Since transient overexpression of β5 did not fully restore the proteasome activity decreased by *THAP1* knockout, possibly due to transfection efficiency and duration of overexpression, we generated a HEK293T cell line stably expressing β5-FLAG and performed *THAP1* knockout. Stable overexpression of β5 almost completely rescued the decrease in proteasome activity and the accumulation of ubiquitinated proteins caused by the absence of THAP1 (Fig. 4c–e and Supplementary Fig. 4c, d). We also observed a decrease in the precursor form of β1 (Fig. 4e, f, fractions 2–10) and β2 (Fig. 4e, f, fractions 10–12) and efficient incorporation of exogenous β5-FLAG proteins into assembled proteasomes (Fig. 4f, fractions 16–26). Taken together, these results indicate that the regulation of

*PSMB5* expression by THAP1 is responsible for maintaining proteasome activity.

### THAP1 binds to the upstream sequence of the *PSMB5* gene

To elucidate the mechanism of transcriptional regulation of *PSMB5* by THAP1, we analyzed the public ChIP-Seq data available on ChIP-Atlas[35]. Peaks of THAP1 binding were observed in genes encoding β5, Rpn7, and Rpn12, with the upstream sequence of the *PSMB5* gene exhibiting a particularly strong binding frequency (Fig. 5a), consistent with the result that THAP1 specifically regulates the expression of *PSMB5*.

A closer look at the ChIP-Seq data upstream of the *PSMB5* gene revealed two peaks before and after the transcription start site and a total of three putative THAP1 binding sequence motifs[36] (NNTNNNGGCAN, N represents any nucleotide) within the 500-bp sequence around the transcription start site (Fig. 5b).

To investigate the importance of this approximately 1-kb region for *PSMB5* expression, we performed a luciferase reporter assay. The addition of the 1-kb sequence upstream of the luciferase cDNA increased the relative luciferase activity (Fig. 5b), and *THAP1* knockout resulted in a decrease in luciferase activity (Fig. 5c), suggesting that THAP1 activates *PSMB5* expression at least via the 1-kb upstream region of the *PSMB5* gene.

To determine whether THAP1 directly binds to the *PSMB5* promoter region, we performed chromatin immunoprecipitation (ChIP) using FLAG antibodies against exogenously expressed FLAG-THAP1. We observed that THAP1 binding to the *PSMB5* promoter region was significantly higher in cells expressing wild-type THAP1 compared to control cells (Fig. 5d). In contrast, the pathological THAP1 point mutation variant (C54Y) did not co-immunoprecipitate with the *PSMB5* promoter region (Fig. 5d). These results indicate that THAP1 directly binds to the *PSMB5* promoter region, whereas the disease-associated THAP1 C54Y mutant fails to bind the *PSMB5* promoter region effectively.

To evaluate the effect of the pathological point mutation or a defect in the DNA-binding domain of THAP1 on proteasome activity and β5 expression, we overexpressed these variants, along with wild-type THAP1, in *THAP1* knockout cells (Fig. 5e, f, and Supplementary Fig. 5). Overexpression of wild-type THAP1 effectively restored proteasome activity and β5 expression. In contrast, neither the pathological C54Y point mutant THAP1 nor the DNA-binding domain-deficient THAP1 were able to do so. These results underscore the pivotal role of the DNA-binding capacity of THAP1 in regulating β5 expression and proteasome activity.

### THAP1 dysfunction leads to proteasome deficiency in neurons and mice

To investigate whether THAP1 similarly regulates proteasome activity in neuronal cells, we performed *THAP1* knockout in SH-SY5Y cells, a human

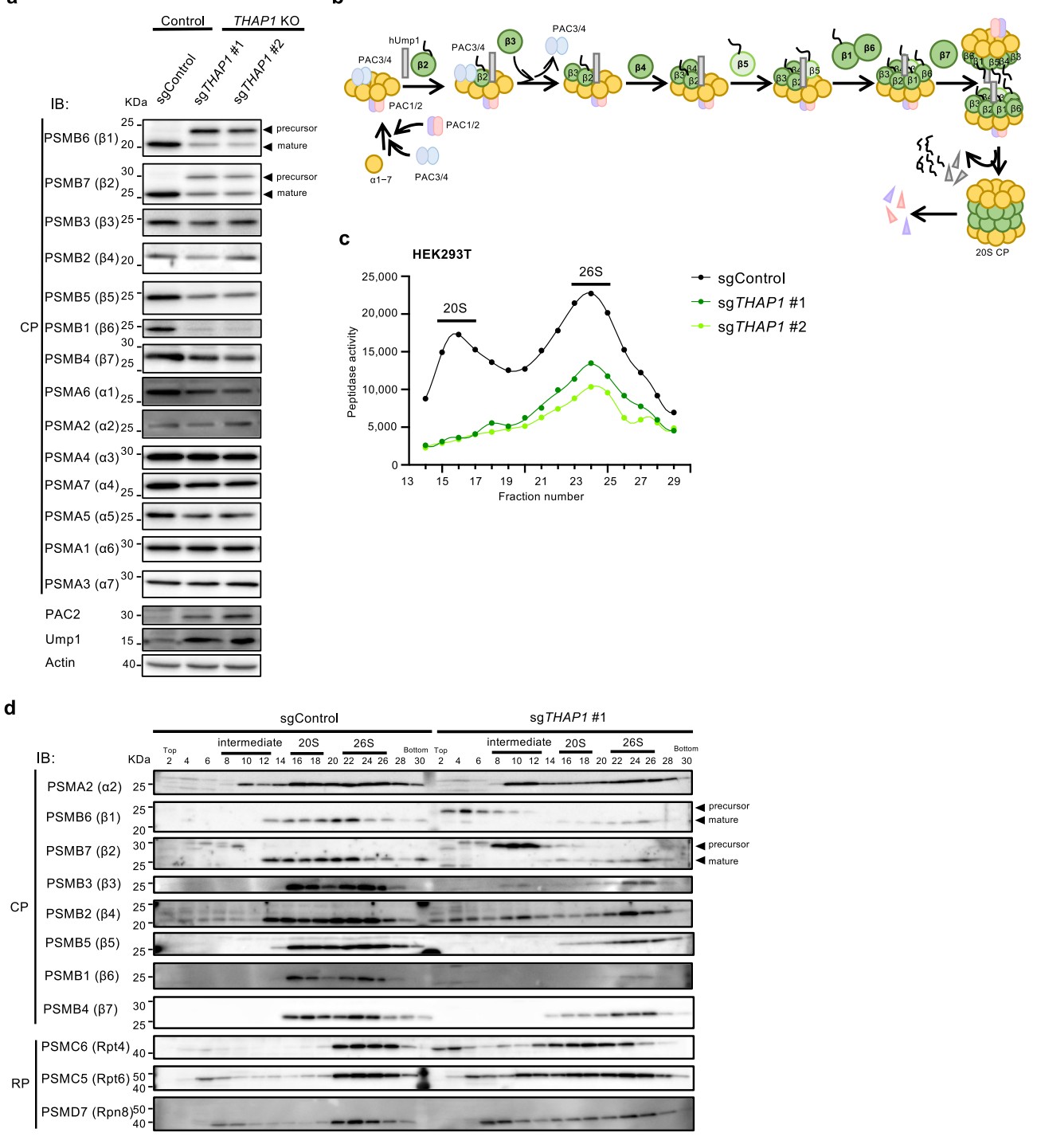

**Fig. 2 | THAP1 is essential for CP assembly. a** Immunoblot analysis of lysates from HEK293T cells using antibodies against the indicated proteins. **b** Schematic diagram of the assembly process of the proteasome CP. **c, d** HEK293T cells were transfected with the indicated sgRNA and selected by puromycin for 6 days. The cell extracts were fractionated by glycerol gradient centrifugation, and an equal amount of each even-numbered fraction was used for measuring proteasome chymotrypsin-like activity in the presence of 0.025% SDS. **c** and for immunoblot analysis using antibodies against the indicated subunits. Fractions 8–12, 16–18, and 22–26 correspond to fractions containing CP assembly intermediates, assembled 20S CP, and the 26S proteasome, respectively (**d**). All experiments were performed at least three biologically independent times with similar results. Source data are provided as a Source Data file.

neuroblastoma-derived cell line. In these cells, we observed a decrease in both β5 mRNA expression and proteasome activity in SH-SY5Y cells (Fig. 6a, b, and Supplementary Fig. 6a). To further support these findings, we analyzed transcriptomic data from previous studies on *Thap1* mutant mice[29], which confirmed reduced *PSMB5* expression in *THAP1* knockout

neuronal tissues (Supplementary Fig. 6b). These results demonstrate that THAP1 also regulates proteasome function in neuronal cells.

Finally, we generated Thap1 C54Y mutant knock-in mice to examine the effects of the dystonia-associated THAP1 mutation on proteasome regulation in vivo. *Thap1^{C54Y/C54Y}* mice were embryonic

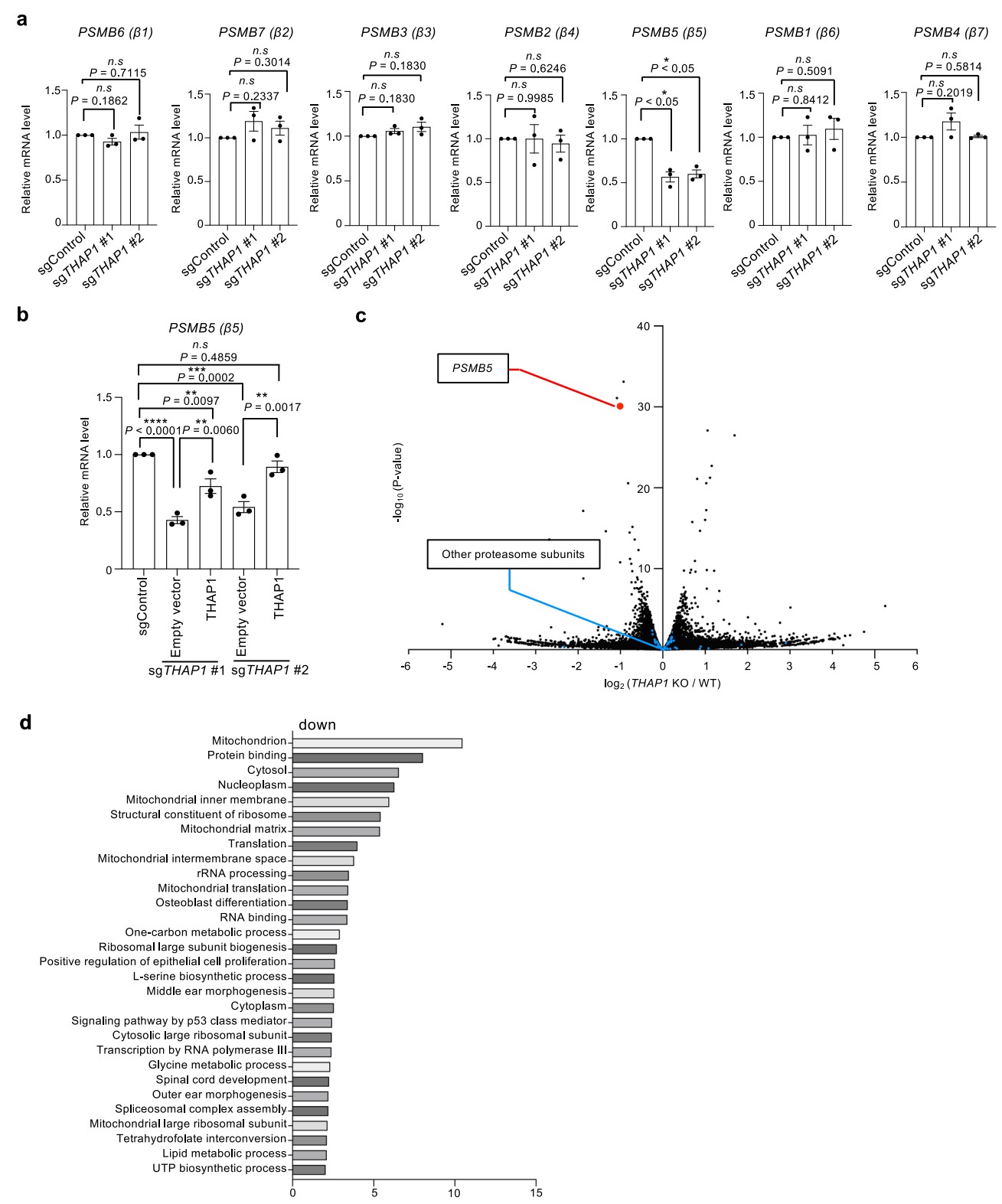

lethal around E10.5, consistent with previous reports[29,37]. RT-qPCR analysis of whole embryos at E10.5 revealed a significant decrease in *Psmb5* transcription in *Thap1^{C54Y/C54Y}* embryos compared to *Thap1^{C54Y/+}* and *Thap1^{+/+}* embryos (Fig. 6c). RNA-seq analysis further confirmed a specific decrease in *Psmb5* expression among proteasome subunits in *Thap1^{C54Y/C54Y}* embryos compared to *Thap1^{+/+}* embryos (Fig. 6d, Supplementary Fig. 6c, and Supplementary Data 2). Biochemical analysis

of embryo lysates showed a marked reduction in both proteasome activity and β5 protein levels in *Thap1^{C54Y/C54Y}* embryos (Fig. 6e, f, and Supplementary Fig. 6d). These results demonstrate that THAP1 regulates proteasome function by mediating β5 transcription in vivo, and that the dystonia-associated mutation disrupts this regulation, raising the possiblity of a link between impaired proteasome activity and the pathogenesis of DYT6 dystonia.

**Fig. 3 | THAP1 regulates the expression of *PSMB5*. a** mRNA expression levels of proteasome β subunit genes in HEK293T cells transfected with the indicated sgRNAs. All data are presented as mean ± SEM (n = 3 from three biological replicates). The significance were calculated using an unpaired two-tailed Student's t-test with Welch's correction. *n.s., not significant*. *$P < 0.05$; **$P < 0.01$; ***$P < 0.001$; ****$P < 0.0001$. **b** mRNA expression level of the β5 subunit gene in *THAP1* knockout cell with or without the addition of sgRNA-resistant *THAP1* cDNA. All data are presented as mean ± SEM (n = 3 from three biological replicates). Statistical significance is indicated by one-way ANOVA Tukey with multiple comparisons test, with exact *P*-values as follows: **$P < 0.01$ ($P_{\text{sgControl vs sg}THAP1 \#1+THAP1} = 0.0097$); ***$P < 0.001$ ($P_{\text{sgControl vs sg}THAP1 \#2} = 0.0002$); *n.s., not significant.* ($P_{\text{sgControl vs sg}THAP1 \#2+THAP1} = 0.4859$); **$P < 0.01$($P_{\text{sg}THAP1 \#1 \text{ vs sg}THAP1 \#1+THAP1} = 0.006$); **$P < 0.01$

($P_{\text{sg}THAP1 \#2 \text{ vs sg}THAP1 \#2+THAP1} = 0.0017$); *n.s., not significant.* *$P < 0.05$; **$P < 0.01$; ***$P < 0.001$; ****$P < 0.0001$. **c** Volcano plot of RNA-seq results of *THAP1* knockout HEK293T cells, compared to control HEK293T cells. HEK293T cells were transfected with a control sgRNA and a *THAP1*-targeting sgRNA, selected by puromycin for 3 days, and subjected to RNA extraction. *PSMB5* and other proteasome subunits are shown in red and blue, respectively. The significance were calculated using an unpaired two-tailed Student's t-test correction. **d** Gene Ontology analysis for biological process of genes downregulated by *THAP1* knockout with the changes of fold change > 0.2 and a *P*-value < 0.01 of (**c**). The significance were calculated using Modified Fisher Exact test and adjusted by Bonferroni, Benjamini, and FDR. All experiments were performed at least three biologically independent times with similar results. Source data are provided as a Source Data file.

## Discussion

In this study, we identified the transcription factor THAP1, the causative gene of hereditary dystonia DYT6, as a proteasome regulator. Of the 33 proteasome subunit genes, THAP1 regulates only the expression of *PSMB5*, which encodes β5 (Fig. 7). The fact that even incomplete loss of THAP1 protein reduced proteasome activity by nearly 50% (Figs. 1 and 2), and that *THAP1* stable knockout cells cannot be established in HEK293T or HCT116 cells (data not shown), indicates that THAP1 plays a crucial role in maintaining proteasome function.

The selective induction of only β5 expression by THAP1 contrasts with the well-known transcription factor Nrf1, which induces the expression of all proteasome subunit genes in response to proteasome dysfunction. Unlike Nrf1, THAP1 functions as a constitutive transcription factor, independent of proteasome dysfunction. The significant decrease in proteasome function caused by the decrease in THAP1 function suggests that the amount of β5 is a rate-limiting factor in proteasome biogenesis under normal conditions. Indeed, overexpression of β5 has been reported to increase proteasome activity in human bone marrow stromal cells, resulting in the maintenance of their multipotency and the suppression of cellular senescence. Therefore, the regulation of β5 expression by THAP1 is a critical mechanism that determines the constitutive expression level of proteasomes. Although THAP1 binding peaks are also observed in the *Rpn7* and *Rpn12* genes in the ChIP atlas (Fig. 5a), it is clear from the RNA-seq analysis of *THAP1* knockout cells and *Thap1^{C54Y/C54Y}* mouse embryos that THAP1 is not involved in the expression of these genes. However, we cannot exclude the possibility that THAP1 might be involved in the expression of *Rpn7* and *Rpn12* under specific conditions.

Dystonia patients exhibit abnormal muscle tension due to excessive brain activity, leading to diverse involuntary movements, abnormal limb positioning, and postural anomalies[38,39]. Maturation of the central nervous system has been shown to be impaired in dystonia patients[40]. To date, 23 loci (DYT1–13, 15–21, 23–25) have been reported to cause hereditary dystonia, and 14 causative genes have been identified; however, the molecular mechanisms underlying the pathogenesis remain elusive. It is believed that DYT6 dystonia results from a loss of THAP1 function, but the specific target gene of THAP1 involved in this disorder remains unclear.

Our findings suggest a potential link between dystonia and proteasomal dysfunction. However, there are many other transcriptional targets of THAP1 besides β5. Gene ontology analysis of genes downregulated in *Thap1^{C54Y/C54Y}* embryos revealed that Thap1 is involved in regulating genes related to the nervous system in mouse development, consistent with previous studies[29,41] (Supplementary Fig. 7). At present, we cannot determine whether proteasome dysfunction is a central factor in the pathogenesis of dystonia. Even if this is the case, it is also unclear how the loss of function of the ubiquitously expressed THAP1 leads to neurological abnormalities. Recent studies have proposed that dystonia, including DYT6, may have a developmental origin, with symptoms manifesting later due to underlying neurodevelopmental defects[38,42]. The identification of THAP1 as a regulator of proteasome function may further support this notion, as proteasome activity plays a crucial role in maintaining cellular homeostasis during development. Given that proteasome activity is essential for processes such as axon guidance, synapse formation, and myelination[43–46], it is plausible that impaired proteasome function during critical developmental windows could contribute to the pathogenesis of DYT6 dystonia. In the future, it will be necessary to clarify the link between dystonia and proteasome dysfunction in DYT6 patients or their mouse models by elucidating whether proteasome function is reduced in specific neurons or whether the phenotype can be rescued by restoring proteasome function through β5 overexpression or other means. Addressing these questions may provide insights and strategies for the treatment of dystonia.

## Methods

### Cell culture and transfection

HEK293T, U2OS, and HeLa cells were purchase from American Type Culture Collection (ATCC) and were cultured in Dulbecco's modified Eagle medium (DMEM, Nacalai Tesque) containing 10% fetal bovine serum (FBS, Thermo Fisher Scientific), 100 U/mL penicillin (Nacalai Tesque), and 100 μg/mL streptomycin (Nacalai Tesque) at 37°C with 5% $CO_2$. SH-SY5Y (ATCC) cells were cultured in Dulbecco's modified Eagle medium (50% DMEM, Nacalai Tesque; 50% Ham's F-12, Nacalai Tesque) and containing 10% fetal bovine serum (FBS, Thermo Fisher Scientific), 100 U/mL penicillin (Nacalai Tesque), and 100 μg/mL streptomycin (Nacalai Tesque) at 37°C with 5% $CO_2$. Transfection was performed using PEI-MAX (MW: 40,000 Da; Polysciences, Warrington). U2OS, HEK293T, HeLa, and SH-SY5Y cells were analyzed 7 days after transfection. Knockout cell lines were established through lentiviral transduction, and cells were selected with 0.5–2.0 μg/mL puromycin.

### Generation of stable cell lines

U2OS cells stably expressing mCherry-P2A-ZsGreen-mODC were generated using a lentiviral conducting system. Lentiviruses were produced by transfecting 293FT cells with AAVS1-based plasmids containing mCherry-P2A-ZsGreen-mODC, as described previously. HEK293T cells stably expressing β5-FLAG were generated using a PEI-MAX transfection.

### Mouse breeding

Mice were maintained in the University of Tokyo, specific pathogen-free animal facility. All animal experimental protocols were approved by the Institutional Animal Care Committee of the Graduate School of Pharmaceutical Sciences, the University of Tokyo (approval number M3-33). C57BL/6 N mice (CLEA Japan) were housed in pathogen-free facilities and maintained in 22 °C, 50% humidity temperature-controlled barrier facilities under a 12–12 h light cycle with access to food and water ad libitum.

### Generation of Thap1 C54Y knock-in mice

Thap1 C54Y knock-in mice were generated using Alt-R™ CRISPR-Cas9 system (IDT). Guide RNA (gRNA) was prepared by annealing AltR-CRISPR-Cas9 tracrRNA and Thap1 crRNA.

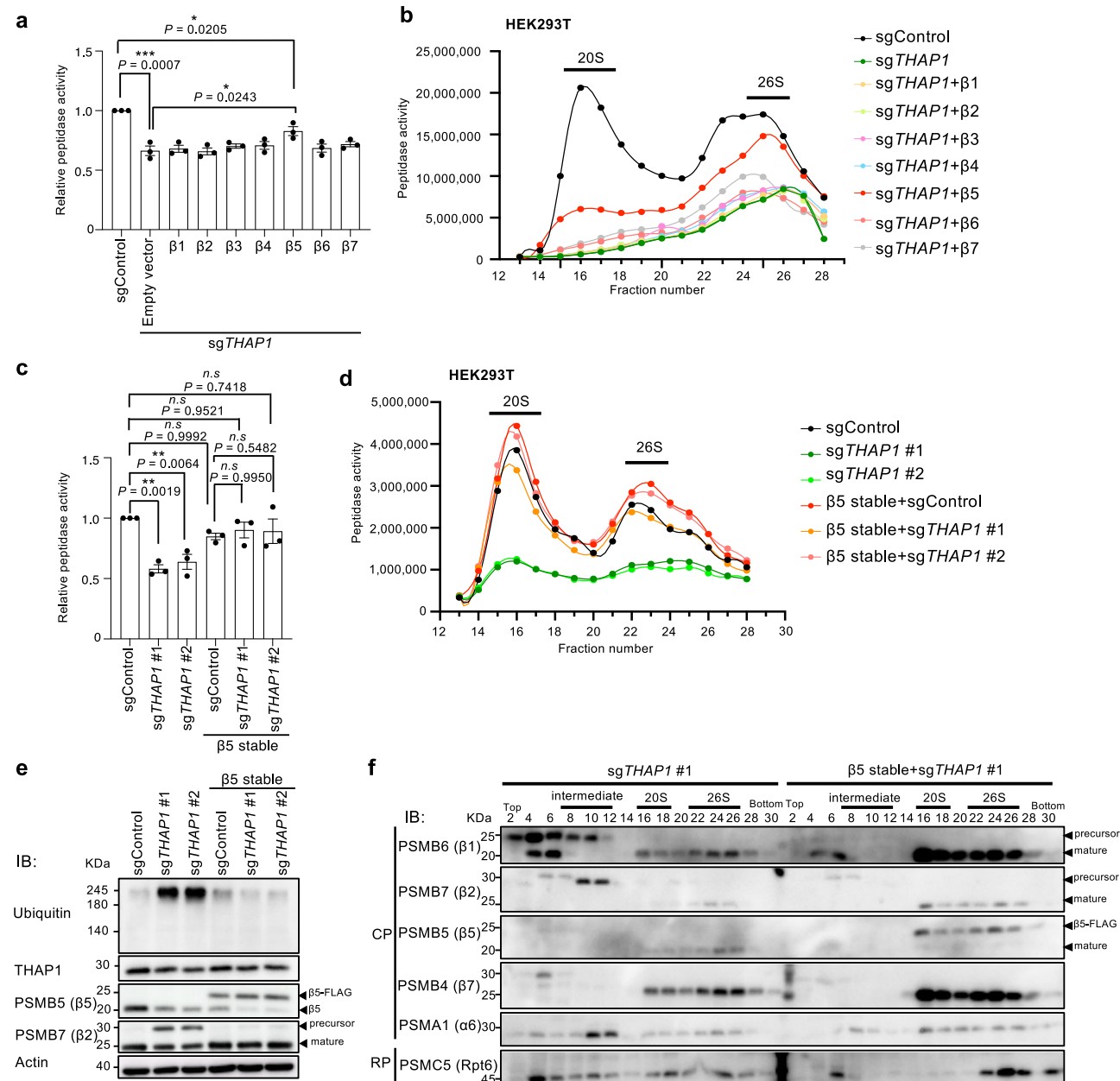

**Fig. 4 | Overexpression of β5 restores proteasome dysfunction caused by THAP1 knockout. a** The HEK293T cells were transfected with the indicated sgRNA targeting *THAP1* and subsequently transfected with each cDNA encoding proteasome β subunits β1–7. After five days, the cells were lysed and subjected to proteasome chymotrypsin-like activity assay. **b** The lysates from (**a**) were fractionated by glycerol gradient centrifugation, and an equal amount of each even-numbered fraction was subjected to a proteasome chymotrypsin-like activity assay. **c, e** The HEK293T cells stably overexpressing β5-FLAG were transfected with sgRNA targeting *THAP1*. Cell lysates were subjected to proteasome chymotrypsin-like activity assay (**c**) and immunoblot analysis using antibodies against the indicated proteins (**e**). **d, f** The lysates from (**c**) were fractionated by glycerol gradient centrifugation, and equal amounts of each even-numbered fraction were subjected to proteasome chymotrypsin-like activity assay. (**d**) and immunoblot analysis using antibodies against the indicated proteins (**f**). Proteasome activity assays were performed in the presence of 0.025% SDS for measuring both the 20S and 26S activities. Data are presented as mean ± SEM (*n* = 3 from three biological replicates). The significance were calculated using an unpaired two-tailed Student's t-test with Welch's correction and one-way ANOVA Tukey with multiple comparisons test. *n.s., not significant.* *P < 0.05; **P < 0.01; ***P < 0.001; ****P < 0.0001. All experiments were performed at least three biologically independent times with similar results. Source data are provided as a Source Data file.

Thap1 crRNA: 5'- GCAGATGCTGCTGTACTTGG - 3', 0.6 μM of gRNA was mixed with 0.3 μM of Guide-it™ Recombinant Cas9 Nuclease (Clontech) to produce ribonucleoprotein complex (RNP). Single strand DNA for Thap1 donor DNA was designed to create XhoI site and to replace 54[th] cysteine with tyrosine (Nihon gene research laboratories). Thap1 C54Y donor ssDNA: 5'-CAGCTGTTAAAAG-GAAAAACTTCAAGCCCACCAAGTACTCGAGCATCTACT -3'. Fertilized zygotes of C57BL/6 N mice were obtained by in vitro

fertilization. Pronuclear stage embryos were introduced with RNP complex and 400 ng/μl donor ssDNA by electroporation using Genome Editor GEB15 (BEX Co. Ltd)[47]. All embryos were cultured overnight in KSOM and the two-cell stage embryos were transferred into the oviducts of pseudopregnant females. Heterozygous offspring (*Thap1*[CS4Y/+]) were identified by genotyping using PCR-based assays on genomic tail DNA. Heterozygous offspring (*Thap1*[CS4Y/+]) were identified by genotyping using PCR on genomic tail DNA.

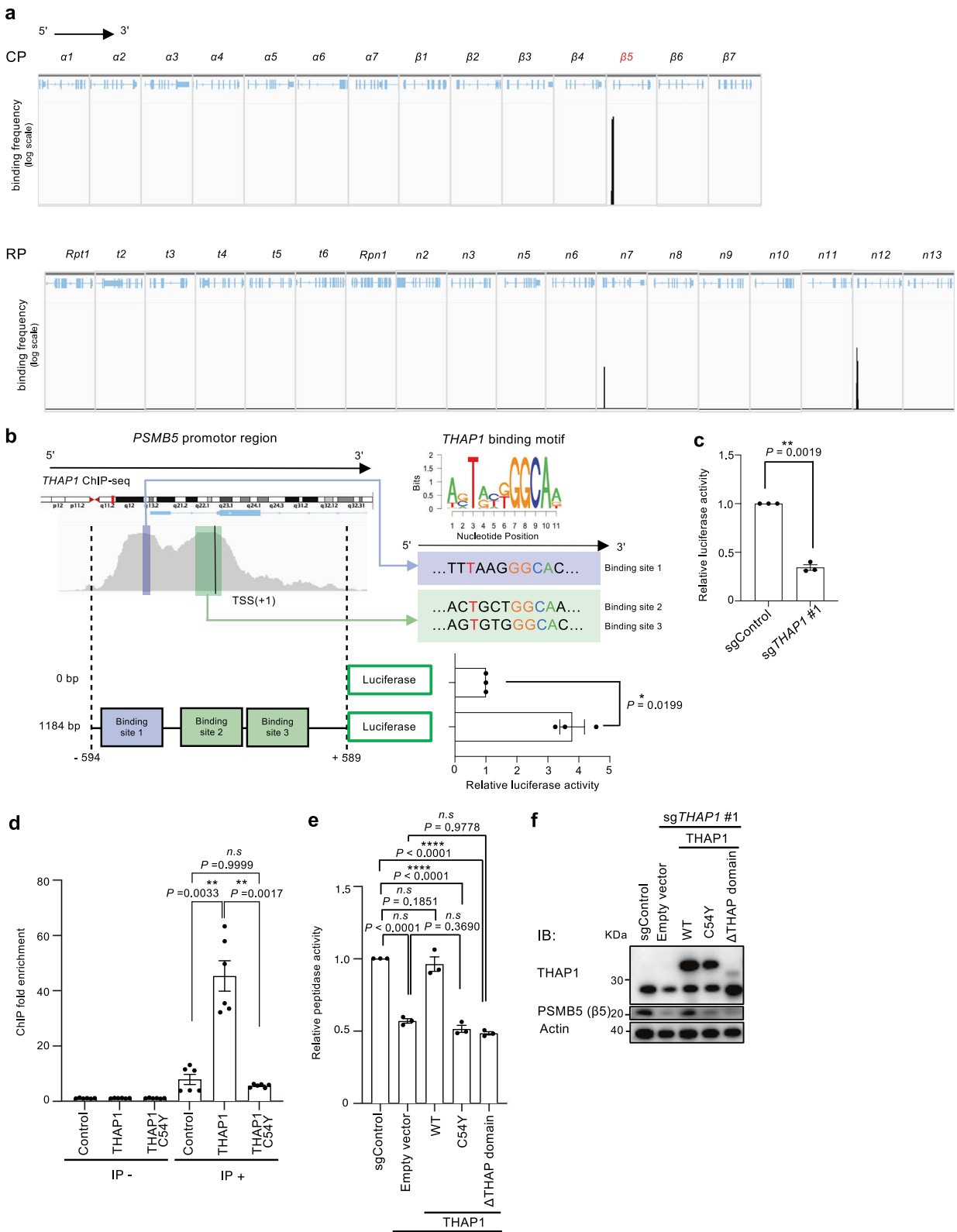

Genomic DNA was used as a template for PCR amplification of the target region with the following primers:

Forward primer: 5′–CACTGCGTCTTTGATATAGAAAAGCACAGG–3′ Reverse primer: 5′–GTAGAATTTGGAAGGTAAAGGCAGGAGGAT–3′.

The PCR products were digested with XhoI restriction enzyme, and the resulting fragments were separated by agarose gel electrophoresis. Genotypes were determined based on the band patterns: wild-type allele (844 bp) and mutant allele (535 bp and 309 bp).

### Genome-wide CRISPR screening

Genome-wide CRISPR screening was performed using ZsGreen accumulation as an index. The Toronto KnockOut CRISPR library - Version 3

**Fig. 5 | THAP1 binds to the upstream sequence of the *PSMB5* gene. a** Analysis of publicly available THAP1 ChIP-seq data with a data height of 120 and data range of 0-2.5. THAP1-binding sites within each proteasome subunit of the CP and RP are shown. **b** Close-up view of THAP1-binding sites on the *PSMB5* gene. Three putative THAP1 binding sites around the transcription start site (TSS) of the *PSMB5* gene were aligned with the THAP1 binding motif sequence and depicted. The constructs and the results of the luciferase assay for promoter activity are shown. Data are presented as mean ± SEM (n = 3 from three biological replicates). The significance were calculated using an unpaired two-tailed Student's t-test with Welch's correction. *P < 0.05. No adjustments were made for multiple comparisons. **c** Relative luciferase activity in HEK293T luciferase reporter cells transfected with the indicated sgRNA. Data are presented as mean ± SEM (*n* = 3 from three biological replicates). The significance were calculated using an unpaired two-tailed Student's t-test with Welch's correction. **P < 0.01. No adjustments were made for multiple comparisons. **d** ChIP-qPCR analysis of THAP1 wild-type and C54Y mutant constructs. Cell lysates from HEK293T cells transfected with cDNAs encoding FLAG-tagged THAP1 wild-type or C54Y mutant cDNA were subjected to ChIP-qPCR to

quantify binding to the promoter region of the *PSMB5* gene. Data are presented as mean ± SEM (n = 3 from three biological replicates). The significance were calculated using two-way ANOVA Tukey with multiple comparisons test. *n.s., not significant*; **P < 0.01. **e, f** Chymotrypsin-like activity (**e**) and immunoblot analysis (**f**) in *THAP1* knockout cells in which sgRNA-resistant THAP1 wild-type, C54Y mutant, and THAP binding-domain deletion were added back. The 20S proteasome activity was measured in the presence of 0.025% SDS. Statistical significance is indicated (**e**) by one-way ANOVA Tukey with multiple comparisons test, with exact *P*-values as follows: *P < 0.0001 ($P_{\text{sgControl vs sg}THAP1\ \#1}$); *n.s., not significant ($P_{\text{sgControl vs sgTHAP1}\ \#1+THAP1}$ = 0.1851); *p < 0.0001 ($P_{\text{sgControl vs sg}THAP1\ \#1+THAP1\ C54Y}$); *P < 0.0001($P_{\text{sgControl vs sg}THAP1\ \#1+THAP1\ \Delta THAP\ domain}$); *n.s., not significant ($P_{\text{sg}THAP1\ \#1\ \text{vs sg}THAP1\ \#1+THAP1\ C54Y}$ = 0.369); *n.s., not significant ($P_{\text{sg}THAP1\ \#1\ \text{vs sg}THAP1\ \#1+THAP1\ \Delta THAP\ domain}$ = 0.9778). Data are presented as mean ± SEM (*n* = 3 from three biological replicates). The significance were calculated using an unpaired two-tailed Student's t-test with Welch's correction. *n.s., not significant*; *P < 0.05; **P < 0.01; ***P < 0.001. All experiments were performed at least three biologically independent times with similar results. Source data are provided as a Source Data file.

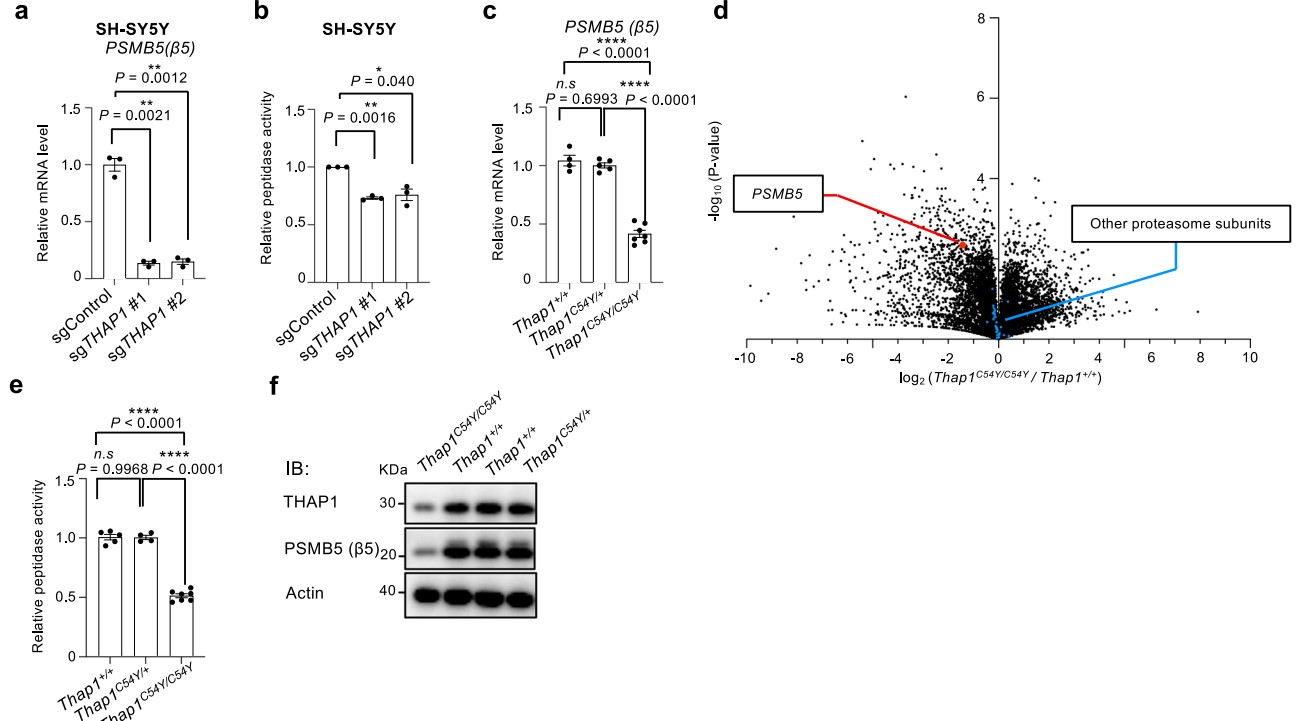

**Fig. 6 | THAP1 regulate proteasome function in neurons and mice embryos. a** mRNA expression levels of proteasome β5 subunit gene in SH-SY5Y cells transfected with the indicated sgRNAs. Data are presented as mean ± SEM (*n* = 3 from three biological replicates). The significance were calculated using an unpaired two-tailed Student's t-test with Welch's correction. **P < 0.01. **b** Proteasome chymotrypsin-like activity of SH-SY5Y cells transfected with indicated sgRNA. The activity was measured in the presence of 0.025% SDS. Data are presented as mean ± SEM (*n* = 3 from three biological replicates). The significance were calculated using an unpaired two-tailed Student's t-test with Welch's correction. *P < 0.05; **P < 0.01. **c** mRNA expression levels of proteasome β5 subunit gene in the *Thap1*[+/+], *Thap1*[C54Y/+] and *Thap1*[C54Y/C54Y] embryos at E10.5. Data represent the mean ± SEM (n = 4 for *Thap1*[+/+] *n* = 5 for *Thap1*[C54Y/+] and n = 6 for *Thap1*[C54Y/C54Y] from three biological replicates). **d** Volcano plot of RNA-seq results of *Thap1*[C54Y/C54Y]

embryos at E10.5, compared to *Thap1*[+/+] embryos at E10.5. *PSMB5* is highlighted in red and other proteasome subunits are colored in blue. Data represent the mean ± SEM (*n* = 3 for *Thap1*[+/+] and n = 3 for *Thap1*[C54Y/C54Y]) The significance were calculated using an unpaired two-tailed Student's t-test correction. **e** Proteasome chymotrypsin-like activity of *Thap1*[C54Y/C54Y] embryos at E10.5. The activity was measured in the presence of 0.025% SDS. **f** The *Thap1*[+/+], *Thap1*[C54Y/+] and *Thap1*[C54Y/C54Y] embryos at E10.5 were subjected to immunoblot analysis with antibodies against the indicated proteins. Data represent the mean ± SEM (*n* = 3 from three biological replicates). Significance was calculated using an unpaired two-tailed Student's t-test with Welch's correction and one-way ANOVA Tukey with multiple comparisons test. *n.s., not significant*; *P < 0.05; **P < 0.01; ***P < 0.001; ****P < 0.0001. All experiments were performed at least three biologically independent times with similar results. Source data are provided as a Source Data file.

(TKO CRISPR library-V3) containing 4 types of sgRNAs targeting 18,053 human genes were used. The TKO CRISPR library-V3 was introduced and amplified by electroporation in Endura electrocompetent cells (Lucigen). TKO CRISPR library-V3 was introduced into $9 \times 10^7$ cells mCherry-P2A-Zsgreen-mODC stable expression U2OS cells by lentivirus, and selection was performed using 2 μg/mL puromycin. The multiplicity of infection (MOI) was measured 4 days after virus

infection. At the same time, the infected cells were split into triplicates. The fluorescence intensity of ZsGreen was measured with SH800 (Sony) 7 days after virus infection, and cells with top 1% ZsGreen were sorted by $1.8 \times 10^6$ cells/replicate. Before sorting, $1.4 \times 10^8$ cells were obtained as a negative control. Genomic DNA was extracted from the obtained cells, and the 360 bp region surrounding the sgRNA was amplified by two-step PCR according to the previous studies[48]. The

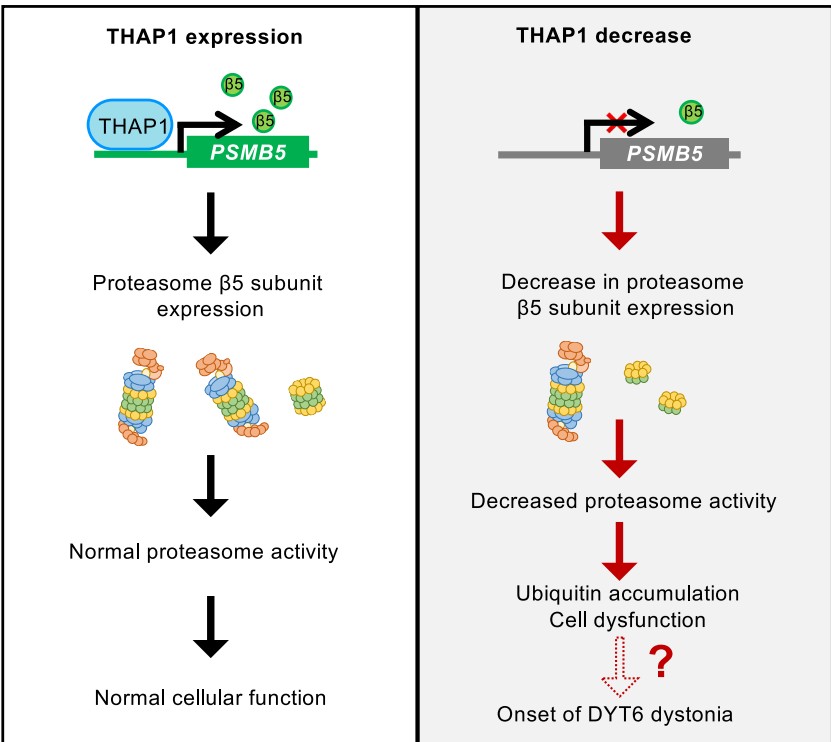

**Fig. 7 | The model of proteasome regulation by THAP1.** THAP1 is responsible for constitutive proteasome activity by regulating the transcription of the *PSMB5* gene. A deficiency in THAP1 activity causes a reduction in *PSMB5* expression, an insufficient production of the β5 subunit, a defect in the assembly of the proteasome CP, and a decline in proteasome activity, which leads to the accumulation of the ubiquitinated proteins and disruption of cellular homeostasis. These results suggest the involvement of proteasome dysfunction due to THAP1 mutations in the development of DYT6 dystonia, but direct evidence is not yet available.

sequences of sgRNAs contained in the obtained PCR products were identified by the next-generation sequencer HiSeq 2500 (Illumina). The RRA score was calculated by MAGeCK-VISPR, and genes with an RRA score of 0.01 or less and a *P*-value of 0.05 or less were identified as hit genes.

### sgRNA expression vector
For the expression of sgRNA and Cas9, the 5′ ends of the two oligo DNAs encoding the sgRNA sequences were phosphorylated, annealed, and then cloned into the BsmBI cleavage site of lentiCRISPR v2. The sequence of sgRNA used are as follows.

Non-targeting sgRNA: 5′ - CTGAAAAAGGAAGGAGTTGA - 3′
*THAP1* sgRNA #1: 5′ - GCGCGCAGGGTCCTCACTTG - 3′
*THAP1* sgRNA #2: 5′ - GTAGCGGTTCTTGCAGCCGT - 3′
*PSMA6* sgRNA: 5′ - GCATACTTACAACAACCAAG - 3′

### Knockout experiment with lentiviral infection of sgRNA
Cells were seeded in 6-well plates at $3.0 \times 10^5$ cells/well. LentiCRISPR v2 with the inserted sgRNA sequence was transfected using PEI-MAX. The medium was changed to DMEM containing 2 μg/mL puromycin for HEK293T, U2OS, and SH-SY5Y cells and 0.5 μg/mL puromycin for HeLa cells 24 hours after transfection, and the cells were cultured at 37 °C for 6 days for drug selection. Unless otherwise stated, samples were collected and analyzed 7 days after transfection.

### Detection of ZsGreen-mODC accumulation by flow cytometry
Cells were harvested after trypsinization, and the fluorescence intensity of ZsGreen and mCherry was analyzed using an Attune NxT Flow Cytometer (ThermoFisher Scientific). The ZsGreen and mCherry fluorescence signals were detected with a 530/30 nm filter and 615/20 nm filter, respectively. Unless otherwise stated, 20,000 events were acquired per sample. Quantitative analysis of fluorescence intensity was performed using Attune NxT Software.

### Glycerol gradient centrifugation analysis and assay of proteasome peptidase activity
Cells were lysed with NP-40 lysis buffer [25 mM Tris-HCl (pH 7.5), 0.2% (v/v) Nonidet P-40, 1 mM dithiothreitol (DTT), 2 mM ATP, and 5 mM $MgCl_2$] and centrifuged at 20,000 g for 20 min at 4 °C. The supernatants were fractionated by 8% to 32% (v/v) linear glycerol gradient centrifugation (22 hours, 83,000 g, 4 °C) and separated into 32 fractions, followed by measurement of peptidase activity and immunoblotting of each fraction, as described previously[49]. Peptidase activities were measured using the fluorescent peptide substrate, succinyl-Leu-Leu-Val-Tyr-7-amido-4-methyl-coumarin for chymotrypsin-like activity; tert-Butoxycarbonyl-Leucyl-Arginyl-Arginyl-4-Methylcoumarin-Amide for trypsin-like activity and Benzyloxycarbonyl-Leucyl-Leucyl-Glutamyl-4-Methylcoumarin-Amide for caspase-like activity, as described previously[50].

### qRT-PCR
For quantitative RT-PCR analysis, total RNA from HEK293T cells was isolated using a High Pure RNA Isolation Kit (Roche) and reverse-transcribed with a ReverTra Ace qPCR RT kit (TOYOBO). Quantitative RT-PCR was performed using Thunderbird Probe qPCR Mix (TOYOBO) and the Universal Probe Library probes (Roche) on the Light Cycler 480 (Roche). The PCR primers used were designed according to the Universal Probe Assay Design Center. The quantitative RT-PCR primer sequences are shown in Supplementary Data 3.

### RNA-seq analysis
For RNA-seq analysis for HEK293T cells, total RNA was extracted from *THAP1* knockout HEK293T cells using a High Pure RNA isolation kit (Roche). RNA-seq libraries were prepared using the NEBNext Ultra II Directional RNA Library Prep Kit (NEB) and sequenced on a DNBSEQ T7 (MGI) to obtain paired-end (PE150) reads. Reads were mapped to Homo Sapiens (GRCh 38/hg38; ensemble 94), using HISAT2[51], and

Feature Counts[52] v1.5.0-p3 was used to count the reads numbers mapped to each gene. Differential expression analysis of two conditions/groups was performed using the DESeq2 R package (1.20.0)[53,54]. The resulting *P*-values were adjusted using the Benjamini and Hochberg approach for controlling the false discovery rate. For RNA-seq analysis for mouse embryo, total RNA was extracted using NucleoSpin RNA kit (TAKARA). Library preparation was performed using FLASH-seq for bulk RNA[55,56]. Sequencing was carried out on an Illumina NextSeq 2000 platform with paired-end 36 bp reads.

### RNA-Seq Data analysis
Sequencing reads were aligned to the UCSC-hg38 provided from Illumina iGenome using STAR (2.7.10) with some options [--outFilterMultimapNmax 20 --alignSJoverhangMin 8 --alignSJDBoverhangMin 1 --outFilterMismatchNmax 999 --alignIntronMin 20 --alignIntronMax 10000 --alignMatesGapMax 1000000]. Row read counts of each gene was calculated using featureCounts (v2.0.1) and wes quantified as TPM. Differentially expressed genes were identified using DESeq2 (v1.20.0). Additional downstream analyses of Gene Ontology (GO) were performed using DAVID[57].

### Immunoblot analysis
Cells were lysed with SDS lysis buffer [50 mM Tris-HCl (pH 8.0), 150 mM NaCl, 0.1% sodium dodecyl sulfate (SDS), 1% Triton X-100, 0.5% sodium deoxycholate, and 1 mM PMSF] and centrifuged at 20,000 *g* for 20 min at 4 °C. The supernatants were collected as cell extracts, which were boiled with SDS sample buffer. Proteins were resolved through SDS-polyacrylamide gel electrophoresis and transferred to Immobilon-P polyvinylidene fluoride membranes (Millipore). The membranes were blocked with Blocking One solution (Nacalai Tesque) and subjected to immunoblotting. Protein bands were visualized using enhanced chemiluminescence detection reagents Western Lightning (PerkinElmer). All images were acquired using a Fusion SL4 system (Vilber Lourmat).

### Antibodies
Antibodies used to identify proteasome subunits were described previously[13,49,58]. The following antibodies were purchased: ubiquitin clone FK2 (AB120; LifeSensors), actin C4 (MAB1501R; Chemicon), FLAG M2 (F1804; sigma), and THAP1 (12584-1-AP; Proteintech).

### Quantitative Chromatin Immunoprecipitation (qChIP)
HEK293T cells were transfected with an empty vector, FLAG-THAP1 expression plasmid, or FLAG-THAP1 C54Y expression plasmid. After 48 hours of transfection, $1 \times 10^7$ cells were fixed in 1% formaldehyde (Nacalai) in DMEM and incubated with 5 μg of anti-FLAG M2 (F1804; Sigma) using M-280 Sheep Dynabeads (Invitrogen) according to the manufacturer's instructions. After washing, elution, and cross-link reversal, DNA was sheared from chromatin (sonicated to 200–500 bp). DNA from each ChIP sample and the corresponding input sample was purified and further analyzed by qPCR as follows: each ChIP sample and a range of dilutions of the corresponding input sample (0.01–2% input) were quantitatively analyzed with gene-specific (*PSMB5* promotor region) primers analyzed with LightCycler 480 (Roche) by THUNDERBRID™ SYBR qPCR mix (TOYOBO). Primers were designed as follows:

| | |
|---|---|
| *PSMB5* ChIP F | CTTCACTTCCTATTAAATCTAT |
| *PSMB5* ChIP R | TTAAGGTTAGTGAAGCCATC |

### Luciferase assay
HEK293T cells were transfected with the plasmid pGL4.21[luc2P/Puro] (Promega), or a plasmid carrying *PSMB5* promoter region and were selected with 2 μg/mL puromycin for 14 days to stably express a reporter gene encoding firefly luciferase. The firefly luciferase activity was measured by GloMax Luminometer (Promega) and normalized by the protein concentration.

### Statistics and reproducibility
All data were derived from at least three independent experiments and are presented as mean ± SEM Unless otherwise stated in the figure legend. Statistical analyses were performed using Microsoft Excel for Mac (v.16.0), R (v.3.6.0), or GraphPad Prism 9.0 software. The appropriate statistical tests, including unpaired two-tailed Student's t-test with Welch's correction, one-way ANOVA with Tukey's multiple comparisons, or two-way ANOVA Tukey with multiple comparisons test were used as indicated in the figure legends. The significance of Gene Ontology (GO) analysis were calculated using Modified Fisher Exact test and adjusted by Bonferroni, Benjamini, and FDR. All experimental replicates yielded consistent results.

### Reporting summary
Further information on research design is available in the Nature Portfolio Reporting Summary linked to this article.

## Data availability
The RNA-seq data comparing control and sgTHAP1 HEK293T generated in this study have been deposited in the DDJB database under accession code E-GEAD-845. Further analysis of the RNA-seq data is provided in Supplementary Data 1. The RNA-seq analysis comparing the E10.5 embryos of mouse control and homozygous C54Y Thap1 mutants generated in this study have been deposited in NCBI's Gene Expression Omnibus (GEO) under accession code GSE:283266 [https://www.ncbi.nlm.nih.gov/geo/query/acc.cgi?acc=gse283266]. Further RNA-seq data analysis is provided in Supplementary Data 2. ChIP-Seq data from the human ChIP-Atlas for THAP1in K562 cells was obtained from NCBI's GEO through accession number GSM803408. Microarray analyses comparing the mouse control and N-CKO for Thap1 was obtained from NCBI's GEO through the accession number GSE97372. Source data are provided with this paper.

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

## Acknowledgements

This work was supported by JSPS KAKENHI (grant no. JP22H00402, JP23H04918 for S.M.), AMED (grant no. 20gm6410008 and 24gm6710028h0001 AMED-PRIME for J.H., 22gm1110003h0006 AMED-CREST for S.M.), and Takeda Science Foundation for S.M. This work was supported by JSPS KAKENHI Grant Number JP 16H06276 (AdAMS).

## Author contributions

Y.W., Yi.W., T.I., E.H., S.H., R.M., and J.H. performed the experiments and analyzed and interpreted the data; Y.W., T.I., E.H., S.H., J.H., and S.M. designed the experiments; A.W., H.S., and Y.G. conducted RNA-Seq analysis for mouse; H.T., and R.Y. generated *Thap1*^C54Y/+ mouse; and Y.W., Yi.W., J.H., and S.M. wrote the manuscript. All authors reviewed the final manuscript and agreed to its publication.

## Competing interests

The authors declare no competing interests.
