## [Transparent Peer Review file · Nature Communications]

The DYT6 dystonia causative protein THAP1 is responsible for proteasome activity via PSMB5 transcriptional regulation

Corresponding Author: Professor Shigeo Murata

Version 0:

Reviewer comments:

Reviewer #1

(Remarks to the Author)

Proteasomal degradation is vital in all cells and organisms, and dysfunction or failure of proteasomal degradation is associated with various human diseases, including cancer and neurodegeneration. Here, the authors have combined genome wide siRNA knockdown and CRISPR knockout screening to identify new genes regulating proteasome function. The screen is based on sorting cells having high levels of the proteasome reporter ZsGreen-mODC, as it accumulates upon proteasome impairment. Using this approach, they identified the transcription factor THAP1 as a regulator of the proteasome, a gene known to be mutated in DYT6 dystonia. They further showed that the loss of THAP1 induces CP assembly defects resulting in decreased proteasome function and the accumulation of ubiquitinated proteins. They found that THAP1 is only regulating the transcription of PSMB5 and that the proteasome defects observed upon THAP1 loss can be partly rescued by PSMB5 overexpression.

On the whole, the paper is well-focused, timely and will bring significant advances to the field of proteasome and neurodegeneration. Data are overall convincing and support the authors' conclusions. The main novelties lie on (1) the discovery that THAP1 selectively controls PSMB5 transcription resulting in proteasome impairment upon its loss, which may be occurring in patients with Dystonia, and (2) that exogenous expression of PSMB5 restores proteasome function in THAP1-deficient cells. In summary, I do think that this paper is a good addition to the proteasome and neurodegeneration fields, and I would support publication of this manuscript providing the more specific concerns listed below are addressed.

Specific points:

1. Fig 1i should be loaded on the same blot to be better comparable.
2. It would be good to confirm that the caspase and trypsin-like activities of the proteasome are similarly affected by sgTHAP1 (likely as it's an assembly defect but the chymotrypsin activity tested here is from beta5 which is the target of THAP1, so testing beta1 and beta2 would be important).
3. Any idea why no precursor accumulates for Beta-5? All incorporated and processed as it becomes rate limiting? This needs to be at least discussed.
4. Homogenise Fig S3 and show individual data points.
5. In Fig 3B, statistics between WT and the two rescue conditions should also be shown.
6. It would be interesting to test proteasome activity of THAP1-deficient cells after rescuing them with mutants THAP1 (no DNA binding (if known) and the disease mutant C54Y mutant).
7. Full excel table with hits up and down in supplementary information would help the readers interrogate the data.
8. No supplementary Fig 4 while mentioned in the text.
9. Fig 4C: why is the level of beta5 not much higher in cell overexpressing Beta5? It should also be discussed why more beta5 precursor is found in cells rescued with beta-5 while this is not the case in WT and sgTHAP1 cells? Loss of stoichiometry?
10. Fig 5a: What about Rpn12 and Rpt5? They seem to have a strong THAP1 peak as for PSMB5. This needs to be discussed.
11. "THAP1-deficient cells cannot be established in HEK293T or HCT116 cells (data not shown)". Deficient is vague here it could be mutation as well as knock-down which is performed by the authors. Do they mean knockout? This needs to be clarified.

Reviewer #2

(Remarks to the Author)

In this manuscript, Wang and colleagues conduct a genome-wide CRISPR/Cas9 screen for regulators of ZsGreen-mODC reporter construct that is unstable in human cells. They identify (all?) proteasome subunits and the transcription factor THAP1 as positive regulators of ZsGreen-mODC degradation. Following up on THAP1, they show that the gene is required for proper proteasome assembly – not because it directly regulates assembly but because it promotes the expression of PSMB5 ($\beta 5$) subunit of the core particle. They show by ChIP-seq that THAP1 binds the PSMB5 promoter, and that by luciferase assays that the PSMB5 promoter is regulated by THAP1.

The experimental results are mostly convincing and the manuscript is well written. I recommend more experiments addressing the mechanistic connection between THAP1 and PSMB5 by e.g. domain deletions and mutagenesis, and a more comprehensive analysis of the effects of THAP1 on the proteome.

Additional suggestions for improving the manuscript:

- 1) I am surprised by the relatively modest rescue of proteasome activity in THAP1 KO cells by PSMB5 overexpression shown in Figure 4. Is this due to the expression level of PSMB5? I didn't see a blot showing the expression level of exogenous vs endogenous PSMB5. This would be important to show. If the expression levels are similar, it will complicate the claim that proteasome activity is solely regulated by PSMB5 via THAP1.
- 2) The authors should acknowledge previous work on THAP1 and its role in regulating the shieldin complex (Shinoda et al. Mol Cell 2021), as this may have important implications for understanding the pathogenesis of THAP1 dystonia.
- 3) Fig 2C: peptide activity -> peptidase activity
- 4) Fig 6 model: Decrease in proteasome $\beta 5$ subunit expression -> $\beta 5$
- 5) Supplementary Figure 1b: multiple typos and grammar errors
- 6) Please use both official gene symbols (e.g. PSMD2/Rpn1) in Supplementary Figure 2a.

Reviewer #3

(Remarks to the Author)

In this study, the authors utilized a genome-wide genetic screen to identify regulators of proteasome activity, and found that a top hit was THAP1, the causative gene of DYT6 dystonia. The screen included siRNA knockdown and CRISPR knockout screening. The authors notably utilized a mammalian system, the U2OS cell line, which is derived from a differentiated epithelial sarcoma, and has an epithelial morphology. Following identification of THAP1, knockout of expression was shown to lead to proteasomal dysfunction via accumulation of a synthetic gene that requires the proteasome for degradation. These knockout cells also displayed decreased proteasome activity in a direct assay. Multiple proteasome subunits were decreased leading to decreased assembly of proteasome subunits, particularly at the $\beta 5$ component. THAP1 was then shown to regulate the transcription PSMB5 and to bind to the upstream portion of the gene, and overexpression of this gene corrected the proteasome defect in the THAP1 KO cells. PSMB5 expression was noted to be decreased in other THAP1 mutant systems already in the literature, including a KO mouse and mouse ESCs. These studies are all well described, performed, and illustrated, and the logical progression of the studies is nicely presented.

The authors themselves raise several issues which are problematic, and which make this study appear somewhat preliminary. None of the findings were demonstrated in a "DYT6" mouse or even in a neuronal cell line, and THAP1 dysfunction is known to sometimes be cell-type specific and/or mutation specific. Mice are available for such studies as are THAP1 cell lines, including human iPSCs, and should be utilized. In fact, the Gene Ontology pathway, proteasome mediated ubiquitin-dependent protein catabolic process, was a top hit in the overlapping DEGs between Thap1^{+/-} and Thap1C54Y striatal tissue in a published RNA seq study, the latter being the mutation assayed in the current report for DNA binding, thus adding interest in the current study. Thus, for a neurologic disease, which does not actually demonstrate neurodegeneration, the link between the proteasome and dystonia should be investigated in a neuronal system and not limited to previously published data only on PSMB5 expression level.

Version 1:

Reviewer comments:

Reviewer #1

(Remarks to the Author)

I thank the authors for their detailed responses to my questions. They have addressed all my concerns, and this is undoubtedly a much-improved version of the manuscript. I have no further comments and fully support its publication.

Reviewer #2

(Remarks to the Author)

The authors have addressed my comments and the manuscript is now ready for publication from my perspective.

Reviewer #3

(Remarks to the Author)

The authors have paid close attention to the previous comments focused on the need to study the role of dystonia genes in general, and in this case, THAP1 in particular, in neuronal cells and structures. Thus, knockdown studies were performed in SY5Y cells, which were unfortunately not differentiated. THAP1 C54Y mice, previously characterized, were re-created, knowing that they would be homozygote lethal. Therefore, only embryonic day 10.5 whole tissue was studied, and not neuronal structures. Having taken these steps, it is surprising that the authors did not differentiate the SY5Y cells and did not look at older mutant mice, albeit of necessity heterozygote, to determine whether indeed the PSMB5 abnormalities are present in more representative neuronal tissue(s). If yes, these studies would greatly strengthen the notion of proteasome involvement in dystonia DYT6. If not, it would be important to discuss DYT6 as a developmental disease, a notion that is gaining traction.

We sincerely thank the reviewers for their interest in our work and for their constructive feedback. All their comments were valuable and helpful in revising and improving our manuscript. We have carefully addressed each point raised by the reviewers and believe that the revised manuscript now better reflects the strength of our findings. A total of 25 new data have been added to the revised manuscript (Figures 1f, 1h, 1i, 4c–f, 5e, 5f, and 6a–f, and Supplementary Fig. 1d, 1e, 4a, 4c, 4d, 5, 6a, 6c, 6d, and 7). The following are our point-by-point responses to the reviewers' comments (*blue italics: reviewers' comments*).

Reviewer #1

Proteasomal degradation is vital in all cells and organisms, and dysfunction or failure of proteasomal degradation is associated with various human diseases, including cancer and neurodegeneration. Here, the authors have combined genome wide siRNA knockdown and CRISPR knockout screening to identify new genes regulating proteasome function. The screen is based on sorting cells having high levels of the proteasome reporter ZsGreen-mODC, as it accumulates upon proteasome impairment. Using this approach, they identified the transcription factor THAP1 as a regulator of the proteasome, a gene known to be mutated in DYT6 dystonia. They further showed that the loss of THAP1 induces CP assembly defects resulting in decreased proteasome function and the accumulation of ubiquitinated proteins. They found that THAP1 is only regulating the transcription of PSMB5 and that the proteasome defects observed upon THAP1 loss can be partly rescued by PSMB5 overexpression.

On the whole, the paper is well-focused, timely and will bring significant advances to the field of proteasome and neurodegeneration. Data are overall convincing and support the authors' conclusions. The main novelties lie on (1) the discovery that THAP1 selectively controls PSMB5 transcription resulting in proteasome impairment upon its loss, which may be occurring in patients with Dystonia, and (2) that exogenous expression of PSMB5 restores proteasome function in THAP1-deficient cells. In summary, I do think that this paper is a good addition to the proteasome and neurodegeneration fields, and I would support publication of this manuscript providing the more specific concerns listed below are addressed.

Response: We deeply appreciate the reviewer's encouraging and positive assessment of our manuscript. Below, we provide detailed responses to each of the specific points raised.

Specific points:

1. Fig 1i should be loaded on the same blot to be better comparable.

Response: We thank the reviewer for this suggestion. As recommended, we reanalyzed the samples used in the original Fig. 1i, performed an immunoblot analysis on a single blot, and confirmed that the rescue of ubiquitinated protein accumulation occurs upon reintroducing THAP1 in *THAP1* knockout cells. The revised results are presented in Fig. 1i.

2. It would be good to confirm that the caspase and trypsin-like activities of the proteasome are similarly affected by sgTHAP1 (likely as it's an assembly defect but the chymotrypsin activity tested here is from beta5 which is the target of THAP1, so testing beta1 and beta2 would be important).

Response: We appreciate the insightful comment. We measured the chymotrypsin-like, trypsin-like, and caspase-like activities in *THAP1* knockout cells and confirmed that all the activities are significantly decreased. The results are now presented in the revised Fig. 1f, h and Supplementary Fig. 1d, e. These results further support our conclusion that proteasome dysfunction in *THAP1*-deficient cells arises from CP assembly defect rather than a malfunction of $\beta 5$ enzymatic activity.

3. Any idea why no precursor accumulates for Beta-5? All incorporated and processed as it becomes rate limiting? This needs to be at least discussed.

Response: We appreciate the comment. $\beta 5$ expression is reduced in *THAP1* knockout, so the assembly intermediate accumulates just prior to $\beta 5$ incorporation. Therefore, there is no $\beta 5$ in the assembly intermediate that accumulates in *THAP1* knockout. The absence of the precursor forms of $\beta 5$ in the accumulated assembly intermediates, nor in the even lighter fractions, suggests that $\beta 5$ expression is rate limiting in *THAP1* knockout cells. This point is now discussed in the revised manuscript (lines 166-168). $\beta 5$ in the 20S and 26S fractions represents proteasomes that were formed before *THAP1* expression was shut down.

4. Homogenise Fig S3 and show individual data points.

Response: As suggested by the reviewer, we have revised Supplementary Fig. 3 to homogenize the results and include individual data points for clarity and consistency.

5. In Fig 3B, statistics between WT and the two rescue conditions should also be shown.

Response: We understand this concern. We performed a statistical analysis of the data shown in the original Fig. 3b and confirmed that the expressions of *PSMB5* gene were significantly rescued in *THAP1* add-back cells as shown in the revised Fig. 3b.

6. It would be interesting to test proteasome activity of THAP1-deficient cells after rescuing them with mutants THAP1 (no DNA binding (if known) and the disease mutant C54Y mutant).

Response: As suggested by the reviewer, we measured the proteasome activity to test the significance of mutant THAP1 proteins including the DNA binding domain-deficient mutant or the disease-related C54Y mutant. The results showed insufficient rescue of proteasome activity by these mutants, highlighting the importance of these regions for THAP1 function. These results are shown in revised Fig. 5e, f and Supplementary Fig. 5a.

7. Full excel table with hits up and down in supplementary information would help the readers interrogate the data.

Response: We appreciate this suggestion. We have added the Excel tables as Supplementary Tables 1 and 2.

8. No supplementary Fig 4 while mentioned in the text.

Response: We apologize for this oversight. We have now included the relevant data as Supplementary Fig. 4a, b to support the results of Fig. 4b.

9. Fig 4C: why is the level of beta5 not much higher in cell overexpressing Beta5? It should also be discussed why more beta5 precursor is found in cells rescued with beta-5 while this is not the case in WT and sgTHAP1 cells? Loss of stoichiometry?

Response: We appreciate the reviewer's critical comments and suggestions. First, we apologize for not describing the experimental condition correctly and in detail regarding the original Fig. 4. In this experiment, we transiently overexpressed $\beta 5$ for five days. This condition was convenient for evaluating the comparison with the overexpression of other β subunits, which nicely suggested that only $\beta 5$ overexpression rescued the absence of THAP1. However, transient overexpression of a proteasome subunit may not be sufficient to fully assess proteasome function: The accumulation of precursor forms of $\beta 5$ in THAP1 knockout cells transiently overexpressing $\beta 5$ in the original Fig. 4 suggests insufficient incorporation of $\beta 5$ into the newly synthesized proteasome within our experimental time course even though exogenous $\beta 5$ was expressed for five days. The low population of transfected cells would also be a cause of add-back insufficiency. Therefore, we generated cells stably expressing $\beta 5$ -FLAG and transduced these cells with sgTHAP1 to more precisely test the rescuing effect of $\beta 5$. In the revised manuscript, we were able to show that the proteasome dysfunction caused by THAP1 knockout was almost completely rescued by stable overexpression of $\beta 5$, without an accumulation of the precursor form of $\beta 5$, as shown in the revised Fig. 4c–f and Supplementary Fig. 4c, d. Thanks to the reviewer's comments, we were able to demonstrate more clearly that the proteasome dysfunction caused by THAP1 deletion is due to

reduced $\beta 5$ expression by employing a more appropriate experimental system. These points are discussed in the revised manuscript (lines 207-216).

10. Fig 5a: What about Rpn12 and Rpt5? They seem to have a strong THAP1 peak as for PSMB5. This needs to be discussed.

Response: We thank the reviewer for pointing this out. We have re-evaluated the ChIP-Atlas data and refined the Fig. 5a to ensure a consistent height scale for the peaks. While significant peaks were observed for *Rpn7* and *Rpn12*, no clear peak was found for *Rpt5*. Using qPCR, we confirmed that the expression of *Rpn7*, *Rpn12*, and *Rpt5* was not significantly decreased in *THAP1* knockout cells (Supplementary Fig. 3). In addition, only *PSMB5* expression was consistently reduced in *Thap1*^{C54Y/C54Y} mouse embryos, as shown in the revised Fig. 6c and Supplementary Fig. 6c. These findings strongly support the specificity of THAP1 in regulating *PSMB5*, although we acknowledge the potential involvement of *Rpn7* and *Rpn12* in specific contexts. This point has been discussed in the revised manuscript (lines 294–298).

11. “THAP1-deficient cells cannot be established in HEK293T or HCT116 cells (data not shown)”. Deficient is vague here it could be mutation as well as knock-down which is performed by the authors. Do they mean knockout? This needs to be clarified.

Response: We appreciate the reviewer’s suggestion for clarity. The term “deficient” was intended to describe cells stably knocked out for the *THAP1* gene. We have revised the manuscript accordingly to clearly state this point (line 281).

Reviewer #2

In this manuscript, Wang and colleagues conduct a genome-wide CRISPR/Cas9 screen for regulators of ZsGreen-mODC reporter construct that is unstable in human cells. They identify (all?) proteasome subunits and the transcription factor THAP1 as positive regulators of ZsGreen-mODC degradation. Following up on THAP1, they show that the gene is required for proper proteasome assembly – not because it directly regulates assembly but because it promotes the expression of PSMB5 ($\beta 5$) subunit of the core particle. They show by ChIP-seq that THAP1 binds the PSMB5 promoter, and that by luciferase assays that the PSMB5 promoter is regulated by THAP1.

The experimental results are mostly convincing (see below) and the manuscript is well written. The results are highly similar to the accompanying manuscript by the Timms group, and I have no doubt that the fundamental discovery of THAP1 regulating PSMB5 is correct. Both papers present essentially the same core findings, although there are some differences in follow-up studies. For example, this manuscript addresses the effect of THAP1 depletion on proteasome assembly and

activity by glycerol gradient centrifugation and in vitro proteasome assays. However, this manuscript does not quite provide the detailed dissection of THAP1/PSMB5 connection and THAP1 function that the other manuscript does. Judging this manuscript on its own, I would hope to see more experiments addressing the mechanistic connection between THAP1 and PSMB5 by e.g. domain deletions and mutagenesis, and a more comprehensive analysis of the effects of THAP1 on the proteome. But I feel that this is unnecessary in the light of the other work. Together, these manuscripts make a convincing case of THAP1 regulating PSMB5, with likely relevance to dystonia caused by THAP1 mutations.

Response: We sincerely thank the reviewer for their supportive and constructive comments. In response to these suggestions, we have further explored the mechanistic link between THAP1 and PSMB5 and clarified the significance of our findings. Below are our responses to the specific points raised.

I have a few suggestions for improving the manuscript.

1) I am surprised by the relatively modest rescue of proteasome activity in THAP1 KO cells by PSMB5 overexpression shown in Figure 4. Is this due to the expression level of PSMB5? I didn't see a blot showing the expression level of exogenous vs endogenous PSMB5. This would be important to show. If the expression levels are similar, it will complicate the claim that proteasome activity is solely regulated by PSMB5 via THAP1.

Response: We appreciate the reviewer's critical comments and suggestions. In this experiment, we transiently overexpressed $\beta 5$ for five days. This condition was convenient for evaluating the comparison with the overexpression of other β subunits, which nicely suggested that only $\beta 5$ overexpression rescued the absence of THAP1. However, transient overexpression of a proteasome subunit may not be sufficient to fully assess proteasome function: The accumulation of precursor forms of $\beta 5$ in THAP1 knockout cells transiently overexpressing $\beta 5$ in the original Fig. 4 suggests insufficient incorporation of $\beta 5$ into the newly synthesized proteasome within our experimental time course even though exogenous $\beta 5$ was expressed for five days. The low population of transfected cells would also be a cause of add-back insufficiency. Therefore, we generated cells stably expressing $\beta 5$ -FLAG and transduced these cells with sgTHAP1 to more precisely test the rescuing effect of $\beta 5$. In the revised manuscript, we were able to show that the proteasome dysfunction caused by THAP1 knockout was almost completely rescued by stable overexpression of $\beta 5$, without an accumulation of the precursor form of $\beta 5$, as shown in the revised Fig. 4c-f and Supplementary Fig. 4c, d. Thanks to the reviewer's comments, we were able to demonstrate more clearly that the proteasome dysfunction caused by THAP1 deletion is due to reduced $\beta 5$ expression by employing a more appropriate experimental system. These points are discussed in the revised manuscript (lines 207-216).

2) *The authors should acknowledge previous work on THAP1 and its role in regulating the shieldin complex (Shinoda et al. Mol Cell 2021), as this may have important implications for understanding the pathogenesis of THAP1 dystonia.*

Response: We appreciate the reviewer bringing this to our attention. We have now cited the study by *Shinoda et al.* in the revised manuscript and included a discussion of its relevance to understanding the role of THAP1 in dystonia (line 92).

3) *Fig 2C: peptide activity -> peptidase activity*

Response: We appreciate for reviewer's careful reading. Accordingly, we correct the terms in the Figure.

4) *Fig 6 model: Decrease in proteasome b5 subunit expression -> β 5*

Response: We agree with the comment. We correct the figures accordingly.

5) *Supplementary Figure 1b: multiple typos and grammar errors*

Response: We sincerely apologize for the errors. Following the reviewer's comment, we have carefully reviewed Supplementary Fig. 1b and corrected all typos and grammatical errors.

6) *Please use both official gene symbols (e.g. PSMD2/Rpn1) in Supplementary Figure 2a.*

Response: We appreciate the suggestion. According to the comments, we have used both official symbols in the revised Figures.

Reviewer #3

In this study, the authors utilized a genome-wide genetic screen to identify regulators of proteasome activity, and found that a top hit was THAP1, the causative gene of DYT6 dystonia. The screen included siRNA knockdown and CRISPR knockout screening. The authors notably utilized a mammalian system, the U2OS cell line, which is derived from a differentiated epithelial sarcoma, and has an epithelial morphology. Following identification of THAP1, knockout of expression was shown to lead to proteasomal dysfunction via accumulation of a synthetic gene that requires the proteasome for degradation. These knockout cells also displayed decreased proteasome activity in a direct assay. Multiple proteasome subunits were decreased leading to decreased assembly of proteasome subunits, particularly at the β 5 component. THAP1 was then shown to regulate the transcription PSMB5 and to bind to the upstream portion of the gene, and overexpression of this gene corrected the proteasome defect in the THAP1 KO cells. PSMB5 expression was noted to be decreased in other THAP1 mutant systems already in the literature, including a KO mouse and

mouse ESCs. These studies are all well described, performed, and illustrated, and the logical progression of the studies is nicely presented.

Response: We sincerely thank the reviewer for their positive evaluation and detailed comments. Below, we address the specific points raised, including additional experiments to strengthen our conclusions.

The authors themselves raise several issues which are problematic, and which make this study appear somewhat preliminary. None of the findings were demonstrated in a “DYT6” mouse or even in a neuronal cell line, and THAP1 dysfunction is known to sometimes be cell-type specific and/or mutation specific. Mice are available for such studies as are THAP1 cell lines, including human iPSCs, and should be utilized.

*In fact, the Gene Ontology pathway, proteasome mediated ubiquitin-dependent protein catabolic process, was a top hit in the overlapping DEGs between *Thap1*^{+/-} and *Thap1*^{C54Y} striatal tissue in a published RNA seq study, the latter being the mutation assayed in the current report for DNA binding, thus adding interest in the current study. Thus, for a neurologic disease, which does not actually demonstrate neurodegeneration, the link between the proteasome and dystonia should be investigated in a neuronal system and not limited to previously published data only on *PSMB5* expression level.*

Response: We fully acknowledge the importance of studying THAP1 function in neuronal contexts and disease-relevant models. In terms of a neuronal cell line, we used the neuroblastoma cell line SH-SY5Y to evaluate the role of THAP1 in *PSMB5* expression and proteasome activity. We observed a significant reduction in *PSMB5* mRNA expression and proteasome activity in SH-SY5Y by sgTHAP1 transduction, as shown in the revised Fig. 6a, b. Regarding the DYT6 context *in vivo*, we generated *Thap1*^{C54Y} knock-in mice to reveal the relationships between *Thap1* and *Psm5* expression *in vivo*. We observed that *Thap1*^{C54Y/C54Y} mice are embryonic lethal around E10.5, consistent with the previous studies by *Ruiz et al.* in Human Molecular Genetics (2015) and *Yellajoshi et al.* in Developmental Cell (2017). Using E10.5 embryos, we performed qPCR and RNA-seq analyses and measured the proteasome activity, confirming the importance of THAP1 in regulating *Psm5* expression and proteasome activity. These results are presented in the revised manuscript (lines 252–274). We believe that these additions provide insights into the neuronal relevance of our findings and their implications for dystonia pathology.